# MTNet: A combined diagnosis algorithm of vessel segmentation and diabetic retinopathy for retinal images

**Ruochen Liu**[1,2], **Song Gao**[1,2]*, **Hengsheng Zhang**[1,2], **Simin Wang**[1,2], **Lun Zhou**[1,2], **Jiaming Liu**[1,2]

**1** Key Laboratory of Earth Exploration and Infomation Techniques (Chengdu University of Technology), Ministry of Education, Chengdu, China, **2** The College of Mechanical and Electrical Engineering, Chengdu University of Technology, Chengdu, China

* gs@cdut.edu.cn

**Data Availability Statement:** All relevant data are within the manuscript. At present, we uploaded our datasets used to the link: https://doi.org/10.6084/m9.figshare.20518335.v1, named MTNet-datasets.rar.

## Abstract

Medical studies have shown that the condition of human retinal vessels may reveal the physiological structure of the relationship between age-related macular degeneration, glaucoma, atherosclerosis, cataracts, diabetic retinopathy, and other ophthalmic diseases and systemic diseases, and their abnormal changes often serve as a diagnostic basis for the severity of the condition. In this paper, we design and implement a deep learning-based algorithm for automatic segmentation of retinal vessel (CSP_UNet). It mainly adopts a U-shaped structure composed of an encoder and a decoder and utilizes a cross-stage local connectivity mechanism, attention mechanism, and multi-scale fusion, which can obtain better segmentation results with limited data set capacity. The experimental results show that compared with several existing classical algorithms, the proposed algorithm has the highest blood vessel intersection ratio on the dataset composed of four retinal fundus images, reaching 0.6674. Then, based on the CSP_UNet and introducing hard parameter sharing in multi-task learning, we innovatively propose a combined diagnosis algorithm vessel segmentation and diabetic retinopathy for retinal images (MTNet). The experiments show that the diagnostic accuracy of the MTNet algorithm is higher than that of the single task, with 0.4% higher vessel segmentation IoU and 5.2% higher diagnostic accuracy of diabetic retinopathy classification.

## Introduction

In retinal fundus images, fundus vessel structure can reflect the characteristics of retinal artery and vein blood vessels, such as thickness, ambiguity, and bending degree, which are important for observing early ocular diseases. Fig 1 shows the retinal image and its segmentation results. An ophthalmologist can analyze and determine a patient's underlying condition and design a treatment plan by observing abnormalities in fundus images. However, only a few thousand ophthalmologists out of only tens of thousands nationwide can recognize medical images. In addition, manual test results are related to physician experience and the large volume of

**Funding:** The author(s) received no specific funding for this work.

**Competing interests:** The authors have declared that no competing interests exist.

manually processed data leads to misdiagnosis, thereby reducing diagnostic accuracy, as well as relatively high diagnostic costs.

## Current status of retinal vessel segmentation research

Among the retinal fundus structures, blood vessels are one of the most important structures, and their clear segmentation images can effectively assist doctors in judging patients' conditions. At present, a large number of scholars have conducted in-depth scientific research on retinal vessel segmentation algorithms, which are divided into two categories: unsupervised and supervised.

The traditional methods based on unsupervised retinal vessel segmentation include classical template matching and vessel tracking. Chaudhuri et al. [1] proposed a method based on template matching, which mainly uses a two-dimensional Gaussian matcher to enhance fundus images. This method has a good effect on some thick and distinct vessels, but a poor effect on some capillaries and cross vessels segmentation. After that, Li et al. [2] proposed an efficient vessel segmentation method based on multiscale matched filtering, which mainly addresses the problems of difficult capillary segmentation, large variation in the width of some fundus vessels, and inconsistent contrast. Zou et al. [3] proposed a vessel segmentation algorithm based on model matching calculation, which can simply predict the centroid location, width, and edge information of some vessels, while an adaptive tracking strategy is used to track each part of the vessel. Yin et al. [4] proposed a probability-based tracking algorithm to segment blood vessels. The algorithm identifies the boundary regions of blood vessels by iteratively using the continuous features and local grayscale information of vessels, and multiple vessel edge points are selected as much as possible during the tracking process. Finally, the Bayesian approach is used to classify and determine the correct boundary points in the vessel structure.

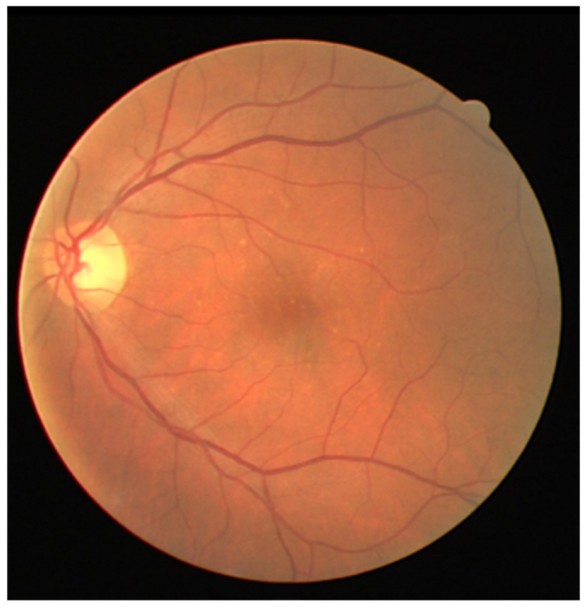

**(a)**

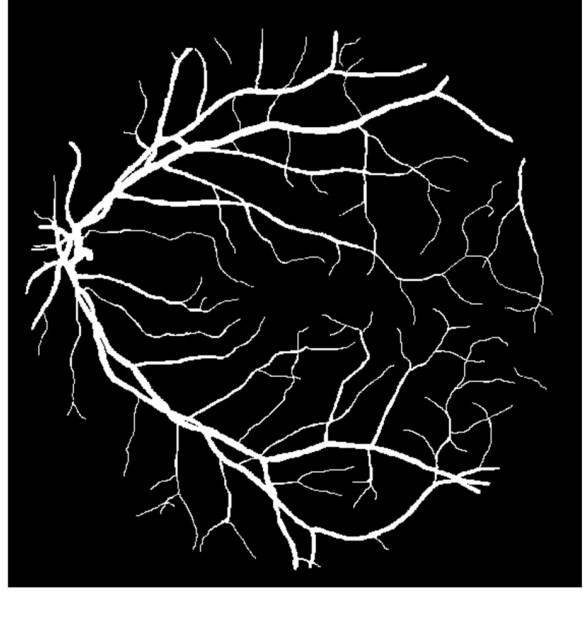

**(b)**

**Fig 1. A retinal vessel segmentation instance.** (a) a retinal image, (b) Corresponding segmentation results.

The supervised retinal vessel segmentation-based algorithm generally requires segmentation label data of vessel images that have been marked in advance by specialized physicians. In the early period of vessel segmentation research, machine learning models based on statistical classification were mainly used, of which three models are represented, namely K-Nearest Neighbor (KNN), Support Vector Machine (SVM), and Extreme Learning Machine (ELA). Staal et al. [5] proposed a supervised clustering segmentation algorithm that regresses the centerline of the blood vessels using a set of linear filters while clustering the nearby pixel points and using the KNN algorithm to use each pixel point as an input to the classifier to make classification judgments to achieve pixel-level segmentation of the retinal blood vessels. Ricci et al. [6] achieved the supervised classification of blood vessels based on two orthogonal vertical division lines after feature extraction using an SVM detector. Zhu et al. [7] proposed an ELM-based vessel segmentation algorithm, which extracts a set of multidimensional feature information vectors for each pixel point, and then uses the vectors as the input of the ELM classifier, and finally, the output of the completed training model corresponds to the segmented retinal vessel image.

With the development of deep learning in the field of artificial intelligence, more and more research hotspots focus on the design of convolutional neural network structures. Fu et al. [8] proposed a segmentation method that views vessel segmentation as a boundary detection task by using a fully convolutional neural network to fuse with a conditional random domain, which in turn generates a vessel probability map. Laskowski et al. [9] proposed a simple neural network algorithm for global contrast normalization and gamma numerical correction of images. Feng et al. [10] proposed a block-based fully convolutional neural network to complete the training of vessel segmentation, which is an algorithm using a jumping convolutional structure with class balance loss and an improved local entropy sampling method. Based on the U-Net network, Fu et al. [11] proposed the M-NET network to study optic disc segmentation and vessel segmentation as a multi-label task. Alom et al. [12] proposed two U-shaped networks modified from U-Net for medical image segmentation, named Recurrent U-Net and R2 U-Net. Cao et al. [13] proposed a network for medical image segmentation named Swin-Unet, an algorithm that feeds shallow image feature information into a Transformer-based U-shaped structure via jump connections for learning local and global semantic feature information. Liu et al. [14] proposed a reverse fusion attention residual network (RFARN), which further improves the continuity and integrity of vessel segmentation by fusing deep local features with shallow global features. The current research on vessel segmentation is mainly focused on the improvement of segmentation accuracy, and the CSP_UNet algorithm proposed in this paper has better segmentation accuracy compared with several classical algorithms.

## Current status of diabetic retinopathy diagnostic research

With increasing interest in diabetic retinopathy, several automated systems for the diagnosis of diabetic retinopathy have been created. When evaluating fundus pictures, ophthalmologists will focus more on the areas of lesions associated with diabetic retinopathy, such as areas of hemorrhage, microaneurysms, soft exudates, and hard exudates, so many researchers have focused their attention on these lesion areas. Shahin et al. [15] extracted pathological feature information with an image processing morphology, such as microaneurysms, soft and hard exudates, and areas of hemorrhage, and then fed these features into a simple neural network structure to implement an automated system for determining diabetic retinopathy. Casanova et al. [16] proposed an algorithm using Random Forest (RF) to distinguish between patients with or without diabetic retinopathy. The two diagnostic methods mentioned above are dichotomous and have achieved good classification results to some extent, but they have not

yet entered into the task of classifying specific lesions with multiple levels of severity and are very limited in clinical application.

With the development of deep learning, many scholars began to explore the use of this method for the detection of diabetic retinopathy and greatly promoted the development of diabetic retinopathy diagnosis. Haloi et al. [17] proposed an algorithm with only five layers of neural networks, which first determines whether each pixel of the image is a microaneurysm and then detects diabetic retinopathy. Alban et al. [18] subjected the images to preprocessing such as denoising, cropping, normalization, and padding, and then migrated them using a pre-trained model for the diabetic retinopathy grading and diagnosis task, and obtained a good classification result. Zhou et al. [19] proposed a deep learning multi-task learning approach using multiple independent network modules to predict the true outcome of fundus images in a classification and regression manner simultaneously. Qomariah et al. [20] used CNN networks for feature extraction and support vector machines instead of SoftMax for image classification. Liu et al. [21] proposed a Swin-Transformer network architecture based on Transformer architecture and achieved good accuracy in various classification, detection, and segmentation tasks. In order to enhance the accuracy of graded diagnosis of diabetic retinopathy, this paper innovatively proposes a joint diagnostic algorithm for fundus vessel segmentation and diabetic retinopathy (MTNet) based on the CSP_UNet algorithm using the hard parameter sharing method in multi-task learning. It is demonstrated experimentally that our algorithm enables a mutually reinforcing effect of the vessel segmentation task and the graded diagnosis task of diabetic retinopathy.

In recent decades, compared to traditional vessel segmentation and diabetic retinopathy diagnosis methods, today's deep learning network-based fundus image diagnosis technology will develop in the direction of automation, precision, serialization, and multifunctional. However, the main research focus of these deep learning-based algorithms is on how to build an effective and powerful segmentation or classification structure, which often requires a large number of computational resources to run and substantial data samples to bring out the excellent performance of these algorithms. This not only ignores the limitations of data sets for various fundus medical diagnostic tasks but also does not consider the limitations of equipment computing power for the deployment of high-performance algorithms in real industrial applications. In addition, the advantages and disadvantages of models based on deep learning for a single diagnostic task versus those for multiple tasks remain to be verified. Overall, it is of great importance to effectively exploit the advantages of medical imaging data in deep learning tasks and to explore the potential of multi-task diagnostic networks in deep learning for medical fundus image diagnosis, as well as to provide an effective technical solution for fundus examination.

The main contributions of our work are as follows:

1. A high-precision automatic segmentation algorithm for the vessel structure of retinal fundus images is proposed, named CSP_UNet. Using four publicly available medical fundus image databases, including DRIVE, STARE, CHASE_DB1, and HRF [22–25], the experimental results reflect the superiority of our algorithm when compared with existing deep learning-based segmentation algorithms in terms of the evaluation metrics such as segmentation accuracy, recall, and algorithm complexity.

2. A combined diagnosis algorithm vessel segmentation and diabetic retinopathy for retinal images was designed in this paper. By introducing a new training method, improving the network structure, and setting the loss function, a complete deep learning model structure is constructed, and its experimental results on the test datasets show that the algorithm achieves higher segmentation accuracy and classification accuracy.

## Methodology

### CSP_UNet

Generally speaking, the feature information located at the bottom layer in the deep network contains more high-frequency features of the image, such as texture, color, edges, etc. However, as the network depth increases, these features are also continuously lost as the network learning progresses, and for segmentation networks, these features cannot be recovered or learned by the network during the upsampling process. Take the U-Net model structure as an example, it adopts the feature map of Encoder and Decoder to connect in a hopping way, so that the underlying high-frequency features and the high-layer features can be better integrated. This effectively ensures that the network can learn features such as image texture, spatial location, and edges while avoiding the higher-level network from learning too many useless features, and the construction of CSP_UNet also draws on this idea. In addition, to further enhance the learning capability of CNNs, the cross-stage network structure CSPNet [26] was introduced to build the underlying network to solve the problem of model inference computation. Also, each module in CSP_UNet's Decoder is connected to a Atrous Spatial Pyramid Pooling (ASPP) [27], which is used to boost the relevance of the vessel category.

Fig 2 is a diagram of the CSP_UNet structure. Similar to the U-Net network structure, CSP_UNet also uses a combination of an Encoder (left side of Fig 2) and a Decoder (right side of Fig 2). First, a convolution of step 1 (Conv1) and convolution of step 2 (Conv2) are used in the head of the Encoder side to complete the feature extraction and resolution adjustment of the input fundus image, followed by three convolution blocks consisting of BottleneckCSP structure and Conv structure serially stacked in the Encoder side.

The BottleneckCSP structure is shown in Fig 3, and consists of two tasks. The structure completes the summation of feature information and the CSP structure completes the

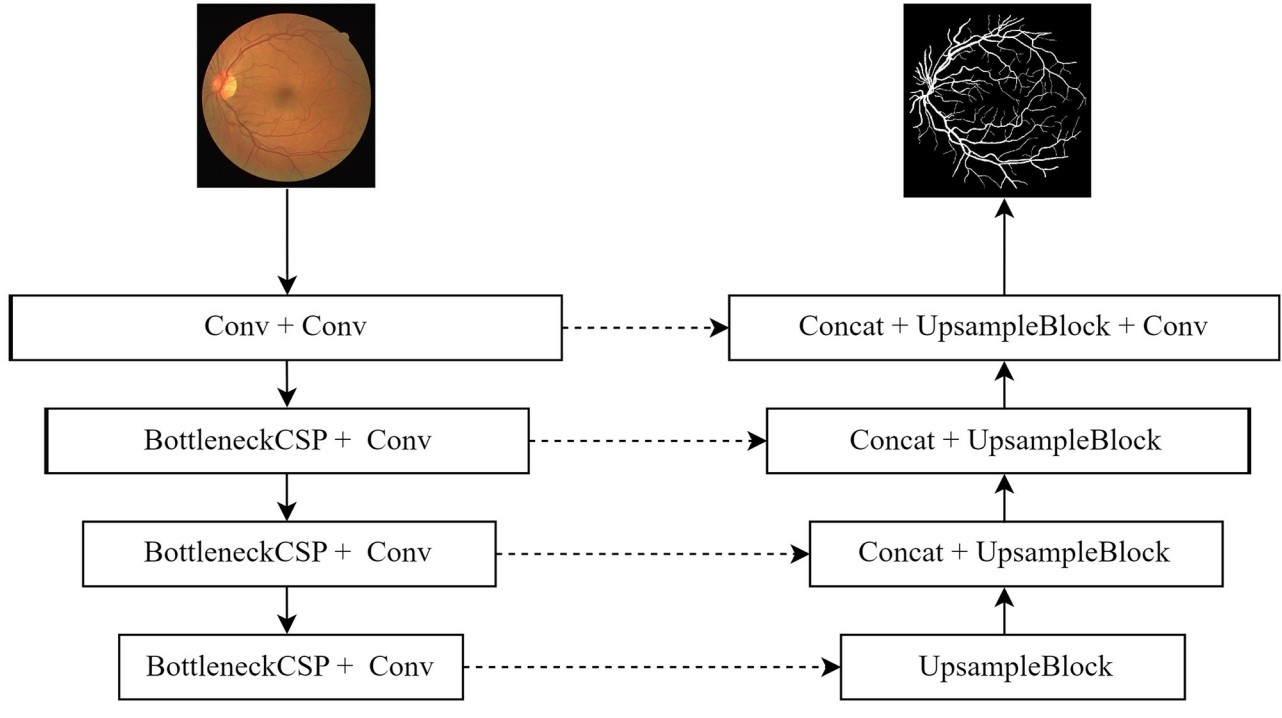

**Fig 2. Schematic diagram of CSP_UNet structure.**

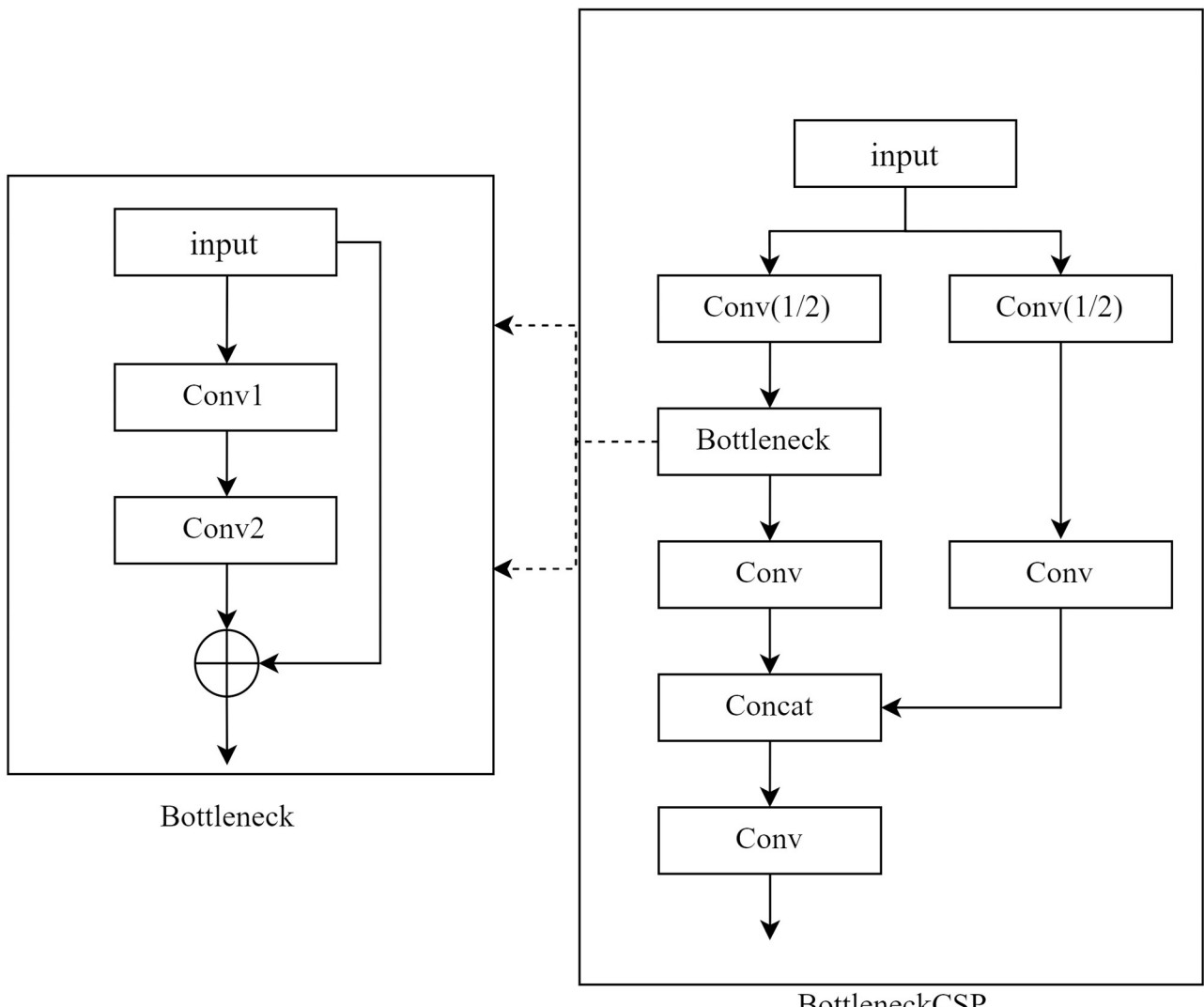

**Fig 3. Schematic diagram of the structure of BottleneckCSP module.**

superposition of channels. These two structures combine the two parts of information from the input features before continuing down the line. First of all, the input of BottleneckCSP goes through the convolution of two contrasting channels Conv(1/2), one side through the Bottleneck structure to complete the summation of features, and another side through a common convolution to complete the superposition of channels with the other side (the Concat structure in Fig 3). The deeper the network, the less the semantic information of the latest generated feature map will be associated with the initial feature map. The deeper feature maps are smaller but retain rich semantic information, while the shallower feature maps are larger in scale and retain basic information such as image structure, color, edges, and location. This is particularly important for learning the details of blood vessels in fundus images to accumulate perceptual features from different domains during the learning process.

In deep learning semantic segmentation tasks, there is the challenge of insufficient correlation between different categories, which is generally solved by simply adding more layers to obtain a larger receptive field, and modifying the convolution to correlate current features

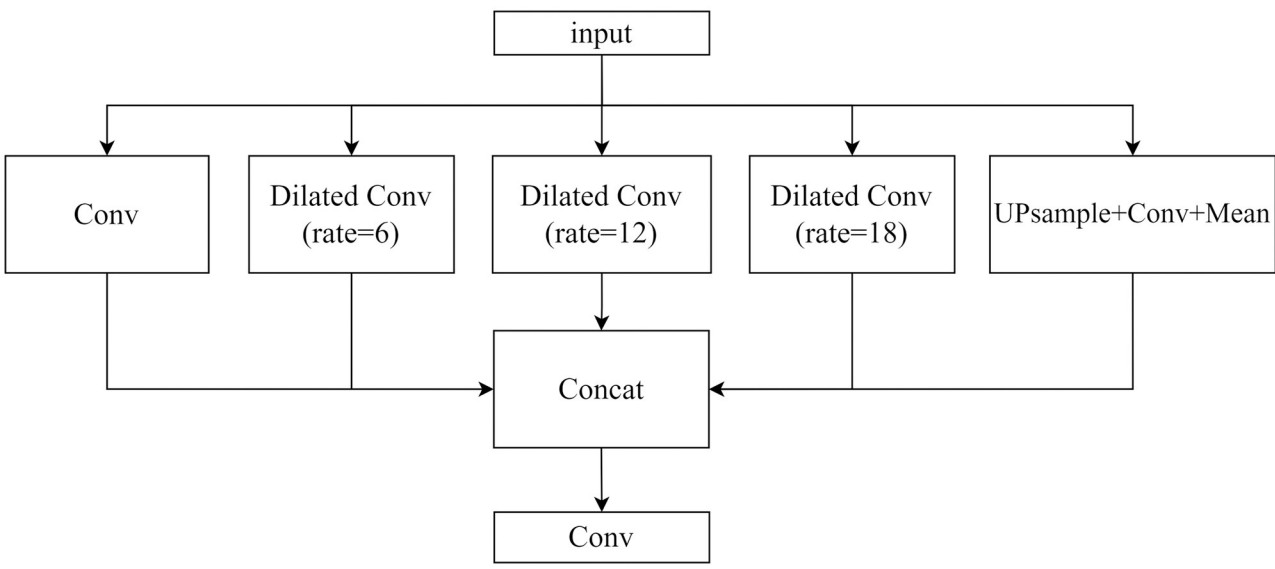

**Fig 4. Schematic diagram of the structure of ASPP module.**

with global features. However, this rough approach tends to cause a steep increase in network computation and slow convergence of the model, and it also requires quite high computing power in the hardware. Thus, the ASPP structure is introduced in each module in the Decoder side of CSP_UNet (Fig 4), and the three dilated convolutions are represented by Conv (rate = 6, 12, 18).

The Decoder side of CSP_UNet is similar to the jump connection structure of U-Net, both mainly through the channel superposition of high-low semantic information, bilinear interpolation up-sampling, convolution to continuously extract, fuse, and recover the resolution of the feature image to obtain the final fused feature map. Corresponding to the Concat module and UpsampleBlock module in Fig 2. CSP_UNet structure diagram, the structure of UpsampleBlock is shown in Fig 5.

In terms of the loss function, the CSP_UNet model uses the Cross Entropy (CE) loss function and the Dice loss function to constrain the model results. The final loss function is shown in Eq (1):

$$L = L_{ce} + L_{dice} \tag{1}$$

## MTNet

In this paper, we use Multi-Task Learning (MTL) to build a single deep learning network (MTNet) to handle the vessel segmentation task and the diabetic retinopathy diagnosis task. MTNet mainly fuses datasets from two different tasks and implements a form of medical fundus dataset augmentation during model training to establish two parallel tasks (segmentation and classification), which improves segmentation accuracy and the diagnostic accuracy of diabetic retinopathy grading in an overall way.

In fact, the vessel segmentation task and the diabetic retinopathy diagnosis task can be analyzed with the same retinal image. The proposed MTNet combined diagnostic algorithm focuses on the analysis of vessel segmentation and diabetic retinopathy diagnosis using two independent parallel branching networks sharing a portion of the parameters. Two datasets,

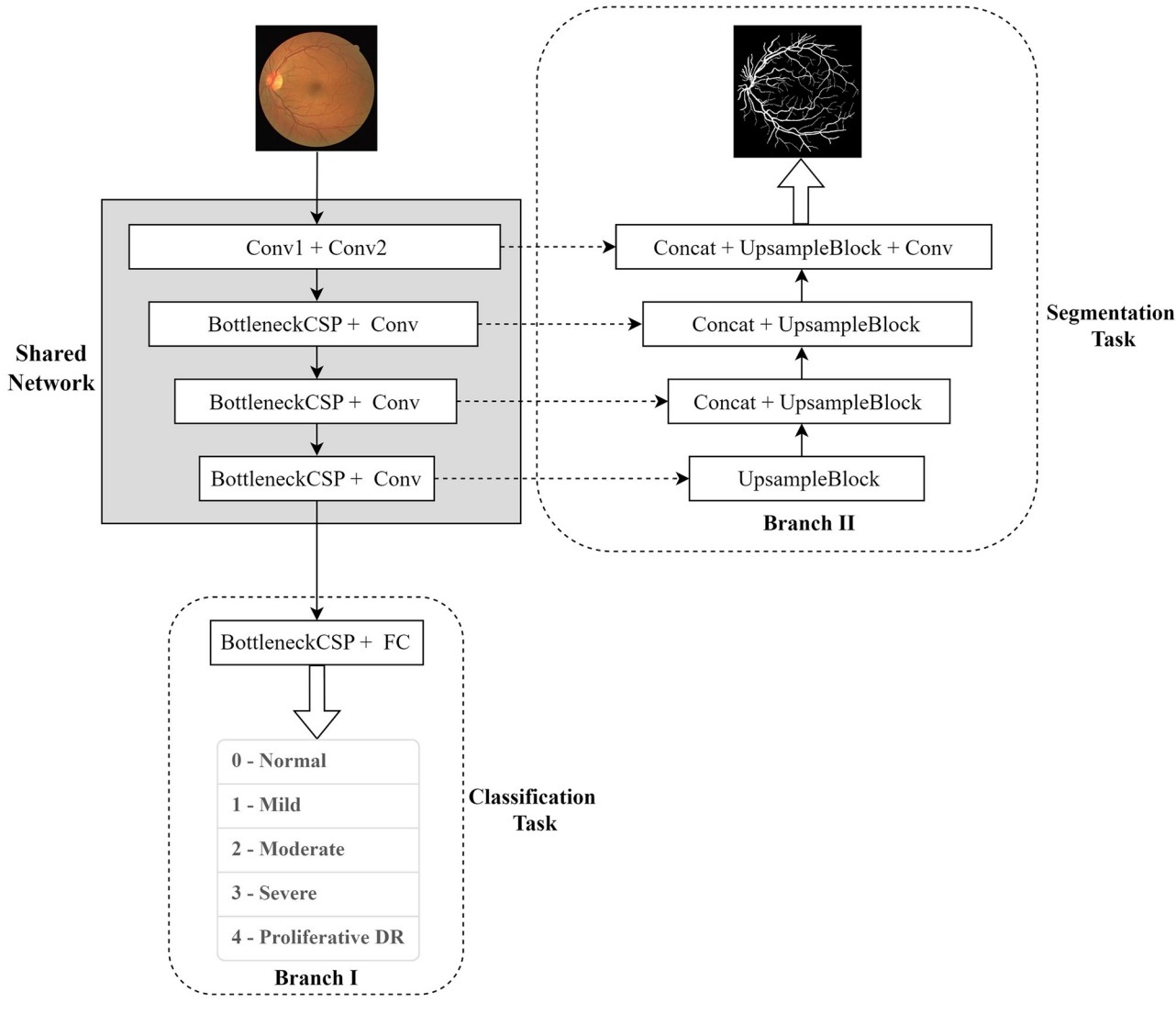

**Fig 5. Structure diagram of UpsampleBlock module.**

vessel segmentation and diabetic retinopathy, were used in the training process. The labels of these two datasets are independent of each other, with one label corresponding to one ophthalmic task in each fundus image, which can effectively alleviate the problem of overfitting during the training process.

Common deep learning convolutional models generally use AlexNet ordinary convolutional structures [28], VGG iterative stacked structures [29], Inception multi-sensory wild convolutional structures [30], network structures composed of ResNet residual units [31], DenseNet densely connected structures [32], and CSPNet cross-stage local network structures and combinations of the above network structures. And MTNet is a multi-task learning strategy based on a hard parameter sharing form, sharing a deep learning base network, whose base network is the backbone network of the CSP_UNet model. In addition, MTNet adds two independent and parallel branches, which are responsible for the retinal vessel segmentation task and the diabetic retinopathy grading diagnosis task respectively. Its cascade network structure diagram is shown in Fig 6.

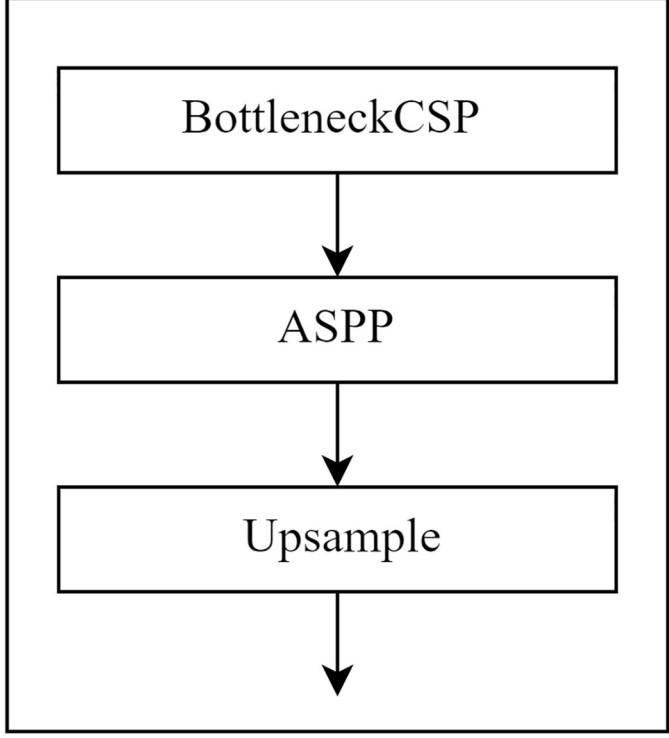

**Fig 6. Structure of the vessel segmentation and diabetic retinopathy diagnostic cascade network.**

The MTNet uses a multi-task learning format, and it is designed to be trained in the following stages.

In the first stage, the diabetic retinopathy data are input, and branch I is trained and classifies the five classes of diabetic retinopathy.

In the second stage, the vessel segmentation data are input, branch II is trained, and outputs the vessel segmentation results in fundus images.

In the third stage, the whole network is jointly trained using diabetic retinopathy data and vessel segmentation data, and after the training is completed, the network parameters of the shared part will be fixed and then the model fine-tuned. Once the model is validated, all parameters will be merged with the MTNet network model. Inputting a fundus image to the model gives direct access to the segmentation and grading diagnostic results for both branches.

In the specific network diagram of the combined diagnosis algorithm, the shared network is the CSPNet network structure mentioned in the previous section, which contains three BottleneckCSP convolutional blocks, and each BottleneckCSP structure consists of multiple convolutional layers, BatchNorm regularization layer, and LeakyReLU activation function. The Fully Connected layer (FC layer) is added to the convolutional block in branch I, and the five probability values are classified by the SoftMax function during model inference. After inference by the Sigmoid function in the last convolution block in branch II, the final vessel segmentation picture is obtained by binarizing the output and judging the output in relation to the threshold of 0.5.

Finally, in terms of loss functions, a cross-entropy loss function was used regarding the diabetic retinopathy diagnosis task as shown in Eq (2), and the loss function shown in Eq (3) was used for the vessel segmentation task. The loss function L of the final MTNet model is a linear sum of two partial loss functions, as shown in Eq (4) below.

$$L(dr)_{ce} = -\sum_{i=1}^{m} y_i log(p_i) \tag{2}$$

$$L(vessel) = \sum_{k=1}^{n} \lambda^k (L_{ce}^k + L_{dice}^k) \tag{3}$$

$$L = L(dr)_{ce} + L(vessel) \tag{4}$$

## Experimental evaluation and results

### Experimental settings

All experiments were using the same hardware platform, a computer with NVIDIA GeForce RTX 3090, Intel(R) Xeon(R) Gold 6252 CPU @ 2.10GHz. All models were built by the PyTorch deep learning framework with the same preprocessing methods and parameter settings. For training, the initial Learning Rate (LR) was set to 1e-3, the Adam optimizer was employed for learning, the learning strategy was Cosine Annealing (CA) decay, the minimum learning rate was 2e-7, the Batch Size (BS) was 4, the number of iterations Epoch was 60. In addition, the model with the minimum loss on the test set is kept at any time during the training process.

### Datasets

In order to verify the performance of the CSP_UNet algorithm, the datasets shown in Table 1 below are selected for controlled experiments to test the performance of the algorithm. The dataset contains four databases, DRIVE, STARE, CHASE_DB1, and HRF.

The training data for MTNet consists of a vessel segmentation dataset and a diabetic retinopathy dataset, where the diabetic retinopathy dataset is derived from data from the Kaggle Aptos2019 competition [33]. It rates the diabetic retinopathy in each image on a scale of 0 to 4: 0-normal, 1-mild, 2-moderate, 3-severe, and 4-proliferative DR, for a total of 3662 images. The experiment randomly selected 2929 images in a ratio of 4:1 as the training set samples and another 733 images as the test set samples, and the distribution of each category is shown in Table 2. And the data source of fundus image and vessel segmentation is the same as that presented in Table 1, with the same fixed same division ratio used to randomly select the training set and the test set.

**Table 1. Distribution of vessel segmentation dataset.**

| Dataset | Total images | Resolution |
|---|---|---|
| DRIVE | 40 | 565×584 |
| STARE | 20 | 605×700 |
| CHASE_DB1 | 28 | 999×960 |
| HRF | 45 | 3504×2336 |

**Table 2. Distribution of diabetic retinopathy data set.**

| Grade of DR | Normal | mild | moderate | severe | Proliferative DR |
|---|---|---|---|---|---|
| Total images | 1805 | 370 | 999 | 193 | 295 |

### Evaluation indicators

To measure the performance of the algorithm comprehensively, this paper uses evaluation metrics including accuracy metrics and complexity metrics. The algorithm was tested using Accuracy, Intersection over Union (IoU), Precision, Recall, and F1-score as our accuracy measures. To evaluate the model complexity, Parameters and Computation (GFLOPs) are used as complexity metrics in the same hardware environment.

In the segmentation task in this paper, the accuracy is the ratio corresponding to the pixels that are successfully predicted among all pixels of the vessel image, as shown in Eq (5).

$$Accuracy = \frac{TP + TN}{TP + FN + TN + FP} \tag{5}$$

The IoU is often used as an important metric in semantic segmentation evaluation. In the vessel segmentation task, the value of IoU is the intersection of the true value of the vessel pixels and the predicted vessel pixels divided by the concatenation, as shown in Eq (6), where R and R′ denote the predicted image result and the actual labeled image respectively and represents the number of 1 in the binary image.

$$IoU = \frac{|R \cap R'|}{|R \cup R'|} \tag{6}$$

Precision is the ratio of pixels that the model predicts to be foreground (vessel portion) or background (without vessel portion) to the true label that is also foreground (vessel portion) or background (without vessel portion). Recall refers to the ratio of pixels predicted by the model as foreground (vessel portion) to those whose true labels are also blood vessels, also known as Sensitivity in semantic segmentation, which has the same meaning as Recall. Unlike the precision and recall rates, the F1-score is used to fully evaluate the precision and recall rates of vessel segmentation and is expressed as a summed average of the two, with a maximum value of 1 and a minimum value of 0, as shown in Eqs (7)–(9).

$$Precision = \frac{TP}{TP + FP} \tag{7}$$

$$Recall = \frac{TP}{TP + FN} \tag{8}$$

$$F1 - score = \frac{2 \times Precision \times Recall}{Precision + Recall} \tag{9}$$

### CSP_UNet algorithm performance test

To test the performance of the CSP_UNet, we selected the vessel segmentation dataset introduced above to train and test it and compared it with some existing deep learning-based vessel segmentation algorithms, including U-Net, AG-Net, DeepLab v3+, R2 U-net, and UperNet [12, 34–37].

**Table 3. Evaluation of accuracy metrics of CSP_UNet with different input sizes.**

| Size | Accuracy | Precision | Recall | F1-score | IoU |
|------|----------|-----------|--------|----------|-----|
| 256×256 | 0.9639 | 0.7882 | 0.9826 | 0.7674 | 0.6219 |
| 512×512 | 0.9683 | 0.8031 | 0.9830 | 0.8006 | 0.6674 |

**Table 4. Complexity measure evaluation of CSP_UNet network model with different input sizes.**

| Dimension | Parameters(MB) | GFLOPs |
|-----------|----------------|--------|
| 256×256 | 15.1 | 12.1 |
| 512×512 | 15.1 | 46.0 |

First, we compared the effect of different retinal fundus image sizes of CSP_UNet. We down-sampled the original retinal images to 256×256 and 512×512 pixels and then performed model training, and the experimental results are shown in Table 3.

From Table 3, it can be obtained that the increase in image size brings different accuracy metrics of vessel segmentation. In particular, there is a large gain in the F1-score and IoU, mainly because the large fundus image preserves more structural details of the vessel region very well.

In addition, to compare the complexity metrics of the models at different input sizes, the number of parameters and computational metrics of the two models were counted, as shown in Table 4. It can be seen that the model consumes less computation when it is fed with 256×256 size fundus images, but as shown by the experimental results in Table 3, there is also a different degree of decrease in each accuracy metric predicted by the model. Therefore, all experiments are completed with input images of 512×512 size.

The segmentation effect of each algorithm is reasonably evaluated on the test dataset based on the above evaluation metrics. The experimental results are shown in Table 5, and the bolded values indicate better performance. Compared with other models, CSP_UNet has an advantage in four evaluation metrics, namely IoU, Accuracy, F1-score and GFLOPs. As the Table 5 shown, the AG-Net model is superior to our model in two metrics, Accuracy and Recall. However, our model has an edge in IoU and F1-score, which is more comprehensive in assessing the performance of vessel segmentation. In terms of computational speed, CSP_UNet also has a strong advantage, due to the fact that the CSP_UNet model uses a large number of cross-stage local network structures instead of ordinary convolutional layers. This structures make the learning capability of network much better and reduce the computational consumption.

Fig 7 shows the results of the two segmentation comparisons for each model on the test set. The first column of the figure represents the input original retinal images, the second column represents the manually labeled real labels, and the third to eighth columns show the

**Table 5. Experimental results of CSP_UNet compared with other segmentation models.**

| Model | IoU | Precision | Recall | F1-score | Accuracy | Parameters(MB) | GFLOPs |
|-------|-----|-----------|--------|----------|----------|----------------|--------|
| U-Net | 0.6640 | 0.8003 | 0.9828 | 0.7981 | 0.9679 | 34.5 | 261.8 |
| AG-Net | 0.6645 | **0.8060** | **0.9835** | 0.7984 | 0.9682 | **9.3** | 65.7 |
| DeepLab v3+ | 0.5664 | 0.7157 | 0.9748 | 0.7240 | 0.9555 | 59.3 | 88.7 |
| R2 U-net | 0.4941 | 0.6254 | 0.9634 | 0.6631 | 0.9429 | 39.1 | 611.2 |
| UperNet | 0.5751 | 0.7260 | 0.9759 | 0.7311 | 0.9569 | 126.0 | 182.1 |
| CSP_UNet | **0.6674** | 0.8030 | 0.9830 | **0.8025** | **0.9683** | 15.1 | **46.0** |

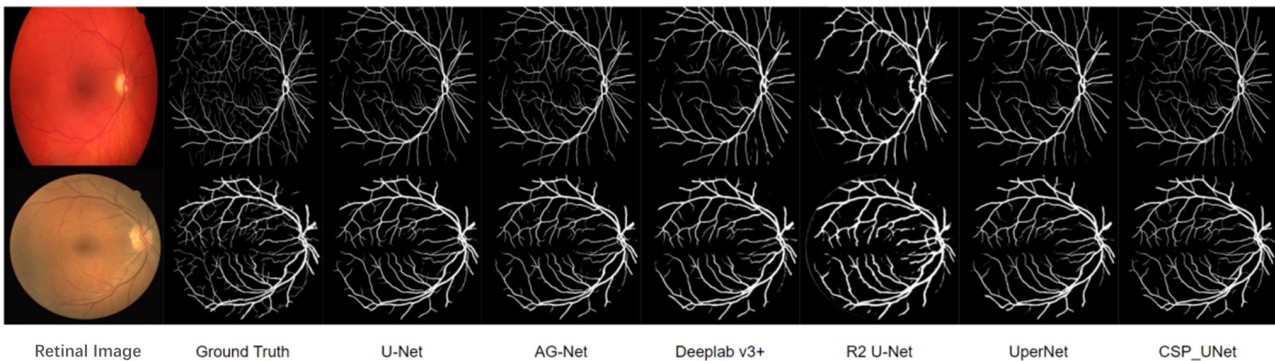

**Fig 7. The segmentation effect of each model on the test set.**

segmentation results of each comparison model. From these figures, CSP_UNet can comprehensively identify the vessel structure of fundus images and has an excellent segmentation effect on the thicker vessel parts. Generally, no intermittent segmentation blocks appear and the lines and points appearing are basically continuous, which indicates that CSP_UNet can effectively predict the specific morphology of the crossed parts of blood vessels. Fig 8 specifically shows the advantages of CSP_UNet with both U-Net and AG-Net models in some segmentation details. The first column shows the real labels, and the last three columns show the image segmentation results of the vessel part of the three models. By comparing the parts circled in blue, we can find that CSP_UNet is more effective in segmenting fine vessels while ensuring the accuracy of coarse vessel segmentation, which is more closely matched to the distribution of real vessels. The red circle shows that CSP_UNet is more sensitive to the segmentation of fine vessels, such as those shown on the right side of the red circle, which U-Net and AG-Net fail to segment, but CSP_UNet is better at preserving these details.

## Test of MTNet combined diagnosis algorithm

Accuracy was used for the assessment of the grading diagnostic, and IoU was employed for the segmentation task. To verify the advantages and disadvantages of the MTNet model based on

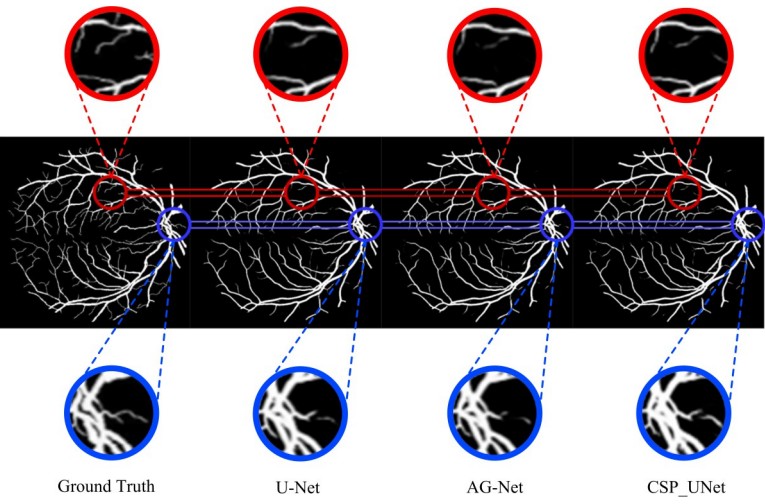

**Fig 8. Comparison of segmentation details of CSP_UNet with U-Net and AG-Net.**

multi-task learning with the single vessel segmentation task model and the single grading diagnostic task model, two groups of single diagnosis tasks were trained as control experiments in the same experimental setting, corresponding to the branch I and branch II has shown in Fig 6 above. The complete training process is specified as follows:

1. Random initialization of the parameters of the two branches and the parameters of the shared network.

2. First, the parameters of branch I (grading diagnostic task) are trained without fixing the parameters of any one network. For the sample labeling of diabetic retinopathy, the diseased samples are sequentially labeled as 0, 1, 2, 3, 4 according to the DR categories, sent to the network for training, and then through the fully connected layer of the branch I. Then the classification probability is obtained after the Sigmoid function, and the graded loss function $L(dr)_{ce}$ is calculated accordingly, after which the network is trained by backpropagation.

3. Stop training when branch I is trained to 60 epochs and save the model with the least loss on the test set.

4. Similar to branch I, branch II is then trained. The network output is updated with parameters by the L(vessel), and training is stopped after 60 epochs, with the training model with the least loss on the test set being saved at any time during the training process.

5. Combine the diabetic retinopathy labeled data with the vessel labeled data. When feeding the network data, input one diabetic retinopathy image followed by one vessel labeled image. After obtaining the output classification probability of branch I and branch II, calculate $L(dr)_{ce}$ and L(vessel) respectively and sum the final loss $L = L(dr)_{ce} + L(vessel)$. Finally, the parameters of the network will be updated by backpropagation, and the two branch networks will share a part of the parameters. The training will stop after 60 epochs and save the model with the least loss on the test set at all times during the process.

6. Fine-tune branch I and branch II, load the shared network of the saved model from the previous stage, and fix this part of the parameters. Then train branch I and branch II, and merge the shared network parameters, branch 1 network parameters, and branch 2 network parameters after training to complete the entire training and preservation of the MTNet.

7. Finally, the trained branch I and branch II as well as the fine-tuned model were experimentally processed and evaluated on the test set for analysis. The results are shown in Table 6.

As can be seen from Table 6, the combined diagnosis algorithm MTNet has a 0.27% improvement in the IoU compared to the single segmentation task with a little difference in computational effort. The MTNet showed a 3.63% improvement in diagnostic accuracy for diabetic retinopathy compared to a single grading diagnostic task. Compared to the number of parameters and computation consumed by the two independent tasks (segmentation task and grading diagnosis task), MTNet requires 26.8% fewer parameters and 31.6% less computation

**Table 6. The experimental results of MTNet.**

| Model | Accuracy | IoU | Parameters(MB) | GFLOPs |
|---|---|---|---|---|
| MTNet(Split task) | — | 0.6674 | **15.1** | 46.0 |
| MTNet(Graded diagnosis task) | 0.6963 | — | 18.8 | **24.9** |
| MTNet | **0.7326** | **0.6701** | 24.8 | 48.5 |

than the other two. The above experimental results demonstrate that the MTNet model has certain advantages over two single-task models.

## Conclusion

In this paper, a deep learning-based algorithm CSP_UNet is designed for the task of retinal vessel segmentation. Compared with five existing classical algorithms, the results show that our algorithm has high segmentation accuracy and better generalization. However, due to the influence of imaging equipment, shooting angle, and illumination, CSP_UNet performs poorly in the face of other retinal image datasets. Considering the fusion of CSP_UNet with unsupervised algorithms may be the key to solving this problem. For the problem of combined diagnosis of retinal vessel segmentation and diabetic retinopathy in retinal images, a deep learning algorithm MTNet is proposed by us, which is capable of learning vessel features and diabetic retinopathy grading information simultaneously. Finally, the experimental results demonstrate that the vessel segmentation task and the graded diagnosis task of diabetic retinopathy have a mutually reinforcing effect. However, further research is needed to determine whether similar findings are available for other medical diagnostic tasks.

## Author Contributions

**Conceptualization:** Ruochen Liu, Hengsheng Zhang.

**Data curation:** Ruochen Liu, Hengsheng Zhang, Lun Zhou.

**Formal analysis:** Ruochen Liu.

**Funding acquisition:** Song Gao.

**Investigation:** Ruochen Liu, Hengsheng Zhang, Lun Zhou.

**Methodology:** Ruochen Liu, Song Gao, Hengsheng Zhang, Simin Wang.

**Project administration:** Ruochen Liu, Song Gao.

**Resources:** Simin Wang, Jiaming Liu.

**Software:** Ruochen Liu, Hengsheng Zhang, Lun Zhou, Jiaming Liu.

**Supervision:** Ruochen Liu, Song Gao, Simin Wang, Jiaming Liu.

**Validation:** Ruochen Liu.

**Visualization:** Ruochen Liu, Song Gao.

**Writing – review & editing:** Ruochen Liu, Song Gao.

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
