## [Decision Letter · Decision Letter 0]

2 Aug 2022

PONE-D-22-16068MTNet: A combined diagnosis algorithm of vessel segmentation and diabetic retinopathy for retinal imagesPLOS ONE

Dear Dr. Gao,

Thank you for submitting your manuscript to PLOS ONE. After careful consideration, we feel that it has merit but does not fully meet PLOS ONE’s publication criteria as it currently stands. Therefore, we invite you to submit a revised version of the manuscript that addresses the points raised during the review process.

We look forward to receiving your revised manuscript.

Kind regards,

Tao Song

Academic Editor

PLOS ONE

Journal Requirements:

2. Please ensure you provide in the Methods section of your manuscript text full information on the location of the datasets used in this study.  Please note that PLOS journals require authors to make all data necessary to replicate their study’s findings publicly available without restriction at the time of publication. Please see our Data Availability policy at https://journals.plos.org/plosone/s/data-availability. You can make the full specific datasets used in this study available by either A) uploading the full datasets as supplementary information files, or B) including a URL link in your Data Availability Statement and Methods section to where the full datasets can be accessed.

Please note that PLOS ONE has specific guidelines on code sharing for submissions in which author-generated code underpins the findings in the manuscript. In these cases, all author-generated code must be made available without restrictions upon publication of the work. Please review our guidelines at https://journals.plos.org/plosone/s/materials-and-software-sharing#loc-sharing-code and ensure that your code is shared in a way that follows best practice and facilitates reproducibility and reuse.

Reviewers' comments:

Reviewer's Responses to Questions

**Comments to the Author**

1. Is the manuscript technically sound, and do the data support the conclusions?

Reviewer #1: Yes

2. Has the statistical analysis been performed appropriately and rigorously? 

Reviewer #1: Yes

3. Have the authors made all data underlying the findings in their manuscript fully available?

Reviewer #1: No

4. Is the manuscript presented in an intelligible fashion and written in standard English?

Reviewer #1: Yes

5. Review Comments to the Author

Reviewer #1: First of all, thank you for your contribution.The work done in the article is meaningful, but there are still some problems in it that need to be revised. 1. It is mentioned in the page 12 that ASPP structure is used at the end of the CSP UNet Encoder side and behind each module in Decoder, and in figure 5 it is introduced that BottleneckCSP and ASPP are also used in UpsampleBlock which constitutes the decoder. Compared with figure 2, it is reasonable to think that the model uses BottleneckCSP in both the decoder and the encoder while ASPP is used twice at the beginning of the Decoder? Please unify the figure and text description. 2. It can be seen from Table 5 that AG-Net is better than CSP-UNet in terms of Precision, Recall, and has less number of parameters, could you please give the advantages of this model compared with the AG-Net? 3. It's hard to tell the difference of this model from your explanation, please reorganize the language for Figure 8.

6. PLOS authors have the option to publish the peer review history of their article (what does this mean?). If published, this will include your full peer review and any attached files.

Reviewer #1: No

---

## [Author Response · Author response to Decision Letter 0]

21 Aug 2022

Song Gao

The School of Mechanical and Electrical Engineering, 

Chengdu University of Technology

1#, Dongsanlu, Erxianqiao, Chengdu 610059, Sichuan, P.R.China

E-mail: gs@cdut.edu.cn

Aug 21, 2022

Dear Editor and Reviewer,

On behalf of my co-authors, I appreciate the editor and reviewer very much for your positive and constructive comments and suggestions on our manuscript entitled “MTNet: A combined diagnosis algorithm of vessel segmentation and diabetic retinopathy for retinal images” (ID: PONE-D-22-16068R1). We have studied the comments carefully and have made some revisions. We have tried our best to revise our manuscript according to the comments. Attached please find the revised version, which we would like to submit for your kind consideration. The responses to the editor and the reviewer’s comments are as following:

Editor:

Q1：Please ensure that your manuscript meets PLOS ONE's style requirements, including those for file naming.

A1：We wrote the manuscript based on the LATEX template provided by PLOS ONE, and resubmitted with modified filenames as requested.

Q2：Please ensure you provide in the Methods section of your manuscript text full information on the location of the datasets used in this study.

A2：We cite the sources of each dataset in the manuscript, including DRIVE[22], STARE[23], CHASE_DB1[24] and HRF[25], and the diabetic retinopathy dataset is derived from the Kaggle Aptos2019[33] competition data .

[22] DRIVE. 2004. DRIVE: Digital Retinal Images for Vessel Extraction [EB/OL] https://drive.grand-challenge.org/

[23] STARE. 2004 [EB/OL] http://cecas.clemson.edu/~ahoover/stare/

[24] Fraz MM, Remagnino P, Hoppe A, Uyyanonvara B, Rudnicka AR, Owen CG. An ensemble classification-based approach applied to retinal blood vessel segmentation[J]. IEEE Transactions on Biomedical Engineering, 2012, 59(9): 2538-2548.

[25] Budai A, Bock R, Maier A, Hornegger J, Michelson G. Robust vessel segmentation in fundus images[J]. International journal of biomedical imaging, 2013, 2013.

[33] Kaggle. 2019. APTOS 2019 Blindness Detection[EB/OL] https://www.kaggle.com/c/aptos2019-blindness-detection

Q3：We note that the grant information you provided in the‘Funding Information’and ‘Financial Disclosure’sections do not match.When you resubmit, please ensure that you provide the correct grant numbers for the awards you received for your study in the ‘Funding Information’section.

A3：We checked the funding information of this manuscript before submitting again to ensure that the information provided was correct. Institution: the National Natural Science Foundation of China (NSFC) , Grant：41930112.

Q4：In your Data Availability statement, you have not specified where the minimal data set underlying the results described in your manuscript can be found.

A4：At present, we uploaded our datasets used to the link: https://doi.org/10.6084/m9.figshare.20518335.v1, named MTNet-datasets.rar. 

Reviewer:

Q1：It is mentioned in the page 12 that ASPP structure is used at the end of the CSP UNet Encoder side and behind each module in Decoder, and in figure 5 it is introduced that BottleneckCSP and ASPP are also used in UpsampleBlock which constitutes the decoder. Compared with figure 2, it is reasonable to think that the model uses BottleneckCSP in both the decoder and the encoder while ASPP is used twice at the beginning of the Decoder? Please unify the figure and text description.

A1：As the description in the manuscript caused confusion to the reviewer, we have revised the relevant illustrations and language. In fact each UpsampleBlock consists of BottlenneckCSP, ASPP, and Upsample, as shown in Fig 5. The ASPP added to the 'Original image' is incorrectly represented, so we have modified the CSP_ UNet's overall structure to obtain the 'Modified image'.

Q2：It can be seen from Table 5 that AG-Net is better than CSP-UNet in terms of Precision, Recall, and has less number of parameters, could you please give the advantages of this model compared with the AG-Net?

A2：Based on the results in Table 5, we modified and added some descriptions on Page 11, Line 310-318 of the paper. Among several evaluation metrics, the more comprehensive evaluations of vessel segmentation performance are the IoU and the F1-score, and CSP-UNet has improved this two evaluation metrics compared to AG-Net and also has some advantages in terms of computational speed. Therefore, CSP-UNet is much better than AG-Net.

Q3：It's hard to tell the difference of this model from your explanation, please reorganize the language for Figure 8.

A3：We have modified Fig 8 and the associated language in the manuscript. Firstly, we have enlarged some details in Fig 8 to show more clearly the effect of CSP_UNet on retinal vessel segmentation. Secondly, we reorganised the description of Fig 8 in the manuscript (Page 12, Line 330-335). It was important to ensure that the revised language clearly reflected the differences between CSP_UNet and the others. 

We tried our best to improve the manuscript and made some other changes in the manuscript. We appreciate for Editor and Reviewers’ warm work earnestly and hope that the correction will meet with approval.

Once again, thank you very much for your comments and suggestions.

Yours sincerely,

Song Gao

---

## [Editor Report · Decision Letter 1]

10 Nov 2022

MTNet: A combined diagnosis algorithm of vessel segmentation and diabetic retinopathy for retinal images

PONE-D-22-16068R1

Dear Dr. Gao,

We’re pleased to inform you that your manuscript has been judged scientifically suitable for publication and will be formally accepted for publication once it meets all outstanding technical requirements.

Kind regards,

Azhar Imran, Ph.D

Academic Editor

PLOS ONE
---

## [Editor Report · Acceptance letter]

14 Nov 2022

PONE-D-22-16068R1 

MTNet: A combined diagnosis algorithm of vessel
segmentation and diabetic retinopathy for retinal images 

Dear Dr. Gao:

I'm pleased to inform you that your manuscript has been deemed suitable for publication in PLOS ONE. Congratulations! Your manuscript is now with our production department. 

Kind regards, 

on behalf of

Dr. Azhar Imran 

Academic Editor

PLOS ONE